# Federated Graph Classification over Non-IID Graphs

**Han Xie, Jing Ma, Li Xiong, Carl Yang**[*]
Department of Computer Science, Emory University
{han.xie, jing.ma, lxiong, j.carlyang}@emory.edu

## Abstract

Federated learning has emerged as an important paradigm for training machine learning models in different domains. For graph-level tasks such as graph classification, graphs can also be regarded as a special type of data samples, which can be collected and stored in separate local systems. Similar to other domains, multiple local systems, each holding a small set of graphs, may benefit from collaboratively training a powerful graph mining model, such as the popular graph neural networks (GNNs). To provide more motivation towards such endeavors, we analyze real-world graphs from different domains to confirm that they indeed share certain graph properties that are statistically significant compared with random graphs. However, we also find that different sets of graphs, even from the same domain or same dataset, are non-IID regarding both graph structures and node features. To handle this, we propose a graph clustered federated learning (GCFL) framework that dynamically finds clusters of local systems based on the gradients of GNNs, and theoretically justify that such clusters can reduce the structure and feature heterogeneity among graphs owned by the local systems. Moreover, we observe the gradients of GNNs to be rather fluctuating in GCFL which impedes high-quality clustering, and design a gradient sequence-based clustering mechanism based on dynamic time warping (GCFL+). Extensive experimental results and in-depth analysis demonstrate the effectiveness of our proposed frameworks.

## 1 Introduction

Federated learning (FL) as a distributed learning paradigm that trains centralized models on decentralized data has attracted much attention recently [28, 53, 25, 18, 17]. FL allows local systems to benefit from each other while keeping their own data private. Especially, for local systems with scarce training data or lack of diverse distributions, FL provides them with the potentiality to leverage the power of data from others, in order to facilitate the performance on their own local tasks. One important problem FL concerns is data distribution heterogeneity, since the decentralized data, collected by different institutes using different methods and aiming at different tasks, are highly likely to follow non-identical distributions. Prior works approach this problem from different aspects, including optimization process [25, 18], personalized FL [13, 6, 8], clustered FL [9, 15, 2], etc.

As more advanced techniques are developed for learning with graph data, using graphs to model and solve real-world problems becomes more popular. One important scenario of graph learning is graph classification, where models such as graph kernels [44, 34, 36, 45] and graph neural networks [21, 43, 49, 46, 47, 48] are used to predict graph-level labels based on the features and structures of graphs. One real scenario of graph classification is molecular property prediction, which is an important task in cheminformatics and AI medicine. In the area of bioinformatics, graph classification can be used to learn the representation of proteins and classify them into enzymes or non-enzymes. For collaboration networks, sub-networks can be classified regarding the information of research areas, topics, genre, etc. More applicable scenarios include geographic networks, temporal networks, etc.

---

[*]Corresponding author.

35th Conference on Neural Information Processing Systems (NeurIPS 2021).

Since the key idea of FL is the sharing of underlying common information, as [23] discusses that real-world graphs preserve many common properties, we become curious about the question, *whether real-world graphs from heterogeneous sources (e.g., different datasets or even divergent domains) can provide useful common information among each other?* To understand this question, we first conduct preliminary data analysis to explore real-world graph properties, and try to find clues about common patterns shared among graphs across datasets. As shown in Table 1, we analyze four typical datasets from different domains, i.e., PTC_MR (molecular structures), ENZYMES (protein structures), IMDB-BINARY (social communities), and MSRC_21 (superpixel networks). We find them to indeed share certain properties that are statistically significant compared to random graphs with the same numbers of nodes and links (generated with the Erdős–Rényi model [7, 10]). Such observations confirm the claim about common patterns underlying real-world graphs, which can largely influence the graph mining models and motivates us to consider the FL of graph classification across datasets and even domains. More details and discussion about Table 1 can be found in Appendix A.

Table 1: Data analysis on important graph properties shared among real-world graphs across different domains. For example, large Kurtosis values [32] indicate long-tail distribution of node degrees, which is observed in ENZYMES, IMDB-BINARY, and MSRC_21; similar average shortest path lengths are observed in PTC_MR, ENZYMES, and MSRC_21, although their actual graph sizes are rather different; large CC are observed in ENZYMES, IMDB-BINARY, and MSRC_21 and large LC are observed in almost all graphs.

| Property | kurtosis of degree distribution | | | avg. shortest path length | | | largest component size (LC, %) | | | clustering coefficient (CC) | | |
|---|---|---|---|---|---|---|---|---|---|---|---|---|
| | real | random | p-value | real | random | p-value | real | random | p-value | real | random | p-value |
| PTC_MR (molecules) | 2.1535 | 2.4424 | 0.9999 | 3.36 | 2.42 | $\sim 0$ | 100 | 82.68 | $\sim 0$ | 0.0095 | 0.1201 | $\sim 0$ |
| ENZYMES (proteins) | 3.0106 | 2.8243 | 0.0027 | 4.44 | 2.56 | $\sim 0$ | 98.24 | 97.69 | 0.2054 | 0.4516 | 0.1425 | $\sim 0$ |
| IMDB-BINARY (social) | 8.9262 | 2.2791 | $\sim 0$ | 1.48 | 1.54 | $\sim 0$ | 100 | 99.93 | 0.0023 | 0.9471 | 0.5187 | $\sim 0$ |
| MSRC_21 (superpixel) | 3.6959 | 2.9714 | $\sim 0$ | 4.09 | 2.81 | $\sim 0$ | 100 | 99.43 | $\sim 0$ | 0.5147 | 0.0655 | $\sim 0$ |

Although common patterns exist among graph datasets, we can still observe certain heterogeneity. In fact, the detailed graph structure distributions and node feature distributions can both diverge due to various reasons. To demonstrate this, we design and evaluate a structure heterogeneity measure and a feature heterogeneity measure in different scenarios (c.f. Section 4.1). We refer to the graphs possibly with significant heterogeneity in our cross-dataset FL setting as non-IID graphs, which concerns both structure non-IID and feature non-IID, where naïve FL algorithms like FedAvg [28] can fail and even backfire (c.f. Section 6.2). Moreover, as the heterogeneity varies from case to case, a dynamic FL algorithm is needed to keep track of such heterogeneity of non-IID graphs while conducting collaborative model training.

Due to the observations that the graphs in one client can be similar to those in some clients but not the others, we get motivated by [2] and find it intuitive to consider a clustered FL framework, which assigns local clients to multiple clusters with less data heterogeneity. To this end, we propose a novel graph-level clustered FL framework (termed GCFL) through integrating the powerful graph neural networks (GNNs) such as GIN [43] into clustered FL, where the server can dynamically cluster the clients based on the gradients of GNN without additional prior knowledge, while collaboratively training multiple GNNs as necessary for homogeneous clusters of clients. We theoretically analyze that the model parameters of GNN indeed reflect the structures and features of graphs, and thus using the gradients of GNN for clustering in principle can yield clusters with reduced heterogeneity of both structures and features. In addition, we conduct empirical analysis to support the motivation of clustered FL across heterogeneous graph datasets in Appendix A.

Although GCFL can theoretically achieve homogeneous clusters, during its training, we observe that the gradients transmitted at each communication round fluctuate a lot (c.f. Section 5.1), which could be caused by the complicated interactions among clients regarding both structure and feature heterogeneity, making the local gradients towards divergent directions. In the vanilla GCFL framework, the server calculates a matrix for clustering only based on the last transmitted gradients, which ignores the client's multi-round behaviors. Therefore, we further propose an improved version of GCFL with gradient-series-based clustering (termed GCFL+).

We conduct extensive experiments with various settings to demonstrate the effectiveness of our frameworks. Moreover, we provide in-depth analysis on the capability of them on reducing both structure and feature heterogeneity of clients through clustering. Lastly, we analyze the convergence of our frameworks. The experimental results show surprisingly positive results brought by our novel setting of cross-dataset/cross-domain FL for graph classification, where our GCFL+ framework can effectively and consistently outperform other straightforward baselines.

## 2 Related works

**Federated Learning** Federated learning (FL) has gained increasing attention as a training paradigm under the setting where data are distributed at remote devices and models are collaboratively trained under the coordination of a central server. `FedAvg` was first proposed by [27] which illustrates the general setting of an FL framework. Since the original `FedAvg` relies on the optimization by SGD, data of non-IID distribution will not guarantee the stochastic gradients to be an unbiased estimation of the full gradients, thus hurting the convergence of FL. In fact, multiple experiments [53, 25, 18] have shown that the convergence will be slow and unstable, and the accuracy will degrade with `FedAvg` when data at each client are statistically heterogeneous (non-IID). [53, 15, 12] proposed different data sharing strategies to tackle the data heterogeneity problem by sharing the local device data or server-side proxy data, which still requires certain public common data, whereas other studies explored the convergence guarantee under the non-IID setting by assuming bounded gradients [39, 51] or additional noise [19]. There are also works seeking to reduce the variance of the clients [26, 18, 25]. Furthermore, multiple works have been proposed to explore the connection between model-agnostic meta-learning (MAML) and personalized federated learning [8, 4]. They aim to learn a generalizable global model and then fine-tune it on local clients, which may still fail when data on local clients are from divergent domains with high heterogeneity. Some personalized FL works [6, 24] studied the bi-level problem of optimization which decouples the local and global optimization, but having each client maintain its own model can lead to high communication cost among the clients and the server. While personalization in the FL setting can address the client heterogeneity problem to some extent, the clustered FL framework [33] can incorporate personalization at the group level to keep the benefits of personalized FL and reduce the communication cost simultaneously.

**Federated Learning on Graphs** Although FL has been intensively studied with Euclidean data such as images, there exist few studies about FL for graph data. [22] first introduced FL on graph data, by regarding each client as a node in a graph. [3] studied the cross-domain heterogeneity problem in FL by leveraging Graph Convolutional Networks (GCNs) to model the interaction between domains. [29] studied spatio-temporal data modeling in the FL setting by leveraging a Graph Neural Network (GNN) based model to capture the spatial relationship among clients. [5] proposed a generalized federated knowledge graph embedding framework that can be applied for multiple knowledge graph embedding algorithms. Moreover, there are several works exploring the GNNs under the FL setting: [16, 54, 40] focused on the privacy issue of federated GNNs; [37] incorporated model-agnostic meta-learning (MAML) into graph FL, which handled non-IID graph data while also preserving the model generalizability; [52] studied the missing neighbor generation problem in the subgraph FL setting; [38] proposed a computationally efficient way of GCN architecture search with FL; [11] implemented an open FL benchmark system for GNNs. Most existing works consider node classification and link prediction on graphs, which cannot be trivially applied to our graph classification setting.

## 3 Preliminaries

### 3.1 Graph Neural Networks (GNNs)

[42] provides a taxonomy that categorizes Graph Neural Networks (GNNs) into recurrent GNNs, convolutional GNNs, graph autoencoders, and spatial-temporal GNNs. In general, given the structure and feature information of a graph $G = (V, E, X)$, where $V$, $E$, $X$ denote nodes, links and node features, GNNs target to learn the representations of graphs, such as a node embedding $h_v \in \mathbb{R}^{d_v}$, or a graph embedding $h_G \in \mathbb{R}^{d_G}$. A GNN typically consists of message propagation and neighborhood aggregation, in which each node iteratively gathers the information propagated by its neighbors, and aggregates them with its own information to update its representation. Generally, an $L$-layer GNN can be formulated as

$$h_v^{(l+1)} = \sigma(h_v^{(l)}, agg(\{h_u^{(l)}; u \in \mathcal{N}_v\})), \forall l \in [L], \tag{1}$$

where $h_v^{(l)}$ is the representation of node $v$ at the $l^{th}$ layer, and $h_v^{(0)} = x_v$ is the node feature. $\mathcal{N}_v$ is neighbors of node $v$, $agg(\cdot)$ is an aggregation function that can vary for different GNN variants, and $\sigma$ represents an activation function.

For a graph-level representation $h_G$, it can be pooled from the representations of all nodes, as

$$h_G = readout(\{h_v; v \in V\}), \tag{2}$$

where $readout(\cdot)$ can be implemented as mean pooling, sum pooling, etc, which essentially aggregates the embeddings of all nodes on the graph into a single embedding vector to achieve tasks like graph classification and regression.

## 3.2 The `FedAvg` algorithm

McMahan et al. [28] proposed an SGD-based aggregating algorithm, `FedAvg`, based on the fact that SGD is widely used and powerful for optimization. `FedAvg` is the first basic FL algorithm and is commonly used as the starting point for more advance FL framework design.

The key idea of `FedAvg` is to aggregate the updated model parameters transmitted from local clients and then re-distribute the averaged parameters back to each client. Specifically, given $m$ clients in total, at each communication round $t$, the server first samples a partition of clients $\{\mathbb{S}_i\}^{(t)}$. For each client $\mathbb{S}_i$ in $\{\mathbb{S}_i\}^{(t)}$, it trains the model downloaded from the server locally with its own data distribution $\mathcal{D}_i$ for $E_{local}$ epochs. The client $\mathbb{S}_i$ then transmits its updated parameters $w_i^{(t)}$ to the server, and the server will aggregate these updates by

$$w^{(t+1)} = \sum_{i=1}^{m} \frac{|D_i|}{|D|} w_i^{(t)}, \tag{3}$$

where $|D_i|$ is the size of data samples in client $\mathbb{S}_i$ and $|D|$ is the total size of samples over all clients. After generating the aggregated parameters (the global model updates), the server broadcasts the new parameters $w^{(t+1)}$ to remote clients, and at the $(t+1)$ round clients use $w^{(t+1)}$ to start their local training for another $E_{local}$ epochs.

# 4 The `GCFL` framework

## 4.1 Non-IID structures and features across clients

From Table 1 we notice that real-world graphs tend to share certain general properties across different graphs, datasets and even domains, which motivates the graph-level FL framework. However, there still exist differences when the detailed graph structures and node features are being considered. In Table 2, we present the average pair-wise structure heterogeneity and feature heterogeneity among graphs in a single dataset, a single domain, and across different domains. Specifically, for structure heterogeneity, we use the Anonymous Walk Embeddings (AWEs) [14] to generate a representation for each graph, and compute the Jensen-Shannon distance between the AWEs of each pair of graphs; for feature heterogeneity, we calculate the empirical distribution of feature similarity between all pairs of linked nodes in each graph, and compute the Jensen-Shannon divergence between the feature similarity distributions of each pair of graphs.

As we can observe in Table 2, both graph structures and features demonstrate different levels of heterogeneity within a single dataset, a single domain, and across different domains. We refer to graphs with such structure and feature heterogeneity as non-IID graphs. Intuitively, directly applying naïve FL algorithms like `FedAvg` on clients with non-IID graphs can be ineffective and even backfiring. To be specific, structure heterogeneity makes it difficult for a model to capture the universally important graph structure patterns across different clients, whereas feature heterogeneity makes it hard for a model to learn the universally appropriate message propagation functions across different clients. How can we leverage the shared graph properties among clients while addressing the non-IID structures and features across clients?

## 4.2 Problem formulation

Motivated by our real graph data analysis in Tables 1 and 2, we propose a novel framework of **G**raph **C**lustered **F**ederated **L**earning (GCFL). The main idea of GCFL is to jointly find clusters of clients with graphs of similar structures and features, and train the graph mining models with `FedAvg` among clients in the same clusters.

Specifically, we are inspired by the Clustered Federated Learning (CFL) framework on Euclidean data [33] and consider a clustered FL setting with one central server and a set of $n$ local clients $\{\mathbb{S}_1, \mathbb{S}_2, \ldots, \mathbb{S}_n\}$. Different from the traditional FL setting, the server can dynamically cluster the

Table 2: Summary of the average heterogeneity of features and structures for some datasets. In general, the structure heterogeneity increases from the settings of one dataset to across-dataset, and to across-domain. However, the feature heterogeneity is more case-by-case, and the high variances indicate that graphs could have large feature divergence even within the same dataset. Additionally, it is not necessarily true that one dataset itself should be more homogeneous (e.g., IMDB-BINARY).

| dataset | IMDB-BINARY (social) | COX2 (molecules) | COX2 (molecules) PTC_MR (molecules) | COX2 (molecules) ENZYMES (proteins) | COX2 (molecules) IMDB-BINARY (social) |
|---|---|---|---|---|---|
| avg. struc. hetero. | 0.4406 ($\pm$0.0397) | 0.3246 ($\pm$0.0145) | 0.3689 ($\pm$0.0540) | 0.5082 ($\pm$0.0399) | 0.6079 ($\pm$0.0331) |
| avg. feat. hetero. | 0.1785 ($\pm$0.1226) | 0.0427 ($\pm$0.0314) | 0.1837 ($\pm$0.1065) | 0.1912 ($\pm$0.1000) | 0.1642 ($\pm$0.1006) |

clients into a set of clusters $\{\mathbb{C}_1, \mathbb{C}_2, \ldots\}$ and maintain $m$ cluster-wise models. In our GCFL setting, each local client $\mathbb{S}_i$ owns a set of graphs $\mathcal{G}_i = \{G_1, G_2, \ldots\}$, where each $G_j = (V_j, E_j, X_j, y_j) \in \mathcal{G}_i$ is a graph data sample with a set of nodes $V_j$, a set of edges $E_j$, node features $X_j$, and a graph class label $y_j$. The task on each local client $\mathbb{S}_i$ is graph classification that predicts the class label $\hat{y}_j = h_k^*(G_j)$ for each graph $G_j \in \mathcal{G}_i$, where $h_k^*$ is the collaboratively learned optimal graph mining model for cluster $\mathbb{C}_k$ to which $\mathbb{S}_i$ belongs. Our goal is to minimize the loss function $F(\Theta_k) := \mathbb{E}_{\mathbb{S}_i \in \mathbb{C}_k}[f(\theta_{k,i}; \mathcal{G}_i)]$, for all clusters $\{\mathbb{C}_k\}$. The function $f(\theta_{k,i}; \mathcal{G}_i)$ is a local loss function for client $\mathbb{S}_i$ which belongs to cluster $\mathbb{C}_k$. In the meantime, we also aim to maintain a dynamic cluster assignment $\Gamma(\mathbb{S}_i) \to \{\mathbb{C}_k\}$ based on the FL process.

## 4.3 Technical design

GNNs are demonstrated to be powerful for learning graph representations and have been wildly used in graph mining. More importantly, the model parameters and their gradients of GNNs can reflect the graph structure and feature information (more details in Section 4.4). Thus, we use GNNs as the graph mining model in our GCFL framework.

Specifically, our GCFL framework can dynamically cluster clients by leveraging their transmitted gradients $\{\Delta\theta_i\}_{i=1}^n$, in order to maximize the collaboration among more homogeneous clients and eliminate the harm from heterogeneous clients. According to [33], if the data distribution of clients are highly heterogeneous, the general FL that trains the clients together cannot jointly optimize all their local loss functions. In this case, after some rounds of communication, the general FL will be close to the stationary point, and the norm of clients' transmitted gradients will not all tend towards zero. Therefore, clustering clients is needed as the general FL approaches to the stationary point. Here, we first introduce a hyper-parameter $\varepsilon_1$ as a criterion to decide whether to stop the general FL based on whether a stationary point is approached, that is,

$$\delta_{mean} = \| \sum_{i \in [n]} \frac{|\mathcal{G}_i|}{|\mathcal{G}|} \Delta\theta_i \| < \varepsilon_1. \tag{4}$$

In the meantime, if there exist some clients still with large norms of transmitted gradients, it means that clients in the group are highly heterogeneous, and thus clustering is needed to eliminate the negative influence among them. We then introduce the second criterion with a hyper-parameter $\varepsilon_2$ to split the clusters when

$$\delta_{max} = \max(\|\Delta\theta_i\|) > \varepsilon_2 > 0. \tag{5}$$

The GCFL framework follows a top-down bi-partitioning mechanism. At each communication round $t$, the server receives $m$ sets of gradients $\{\{\Delta\theta_{i_1}\}, \{\Delta\theta_{i_2}\}, \ldots, \{\Delta\theta_{i_m}\}\}$ from clients in clusters $\{\mathbb{C}_1, \mathbb{C}_2, \ldots, \mathbb{C}_m\}$. For a cluster $\mathbb{C}_k$, if $\delta_{mean}^k$ and $\delta_{max}^k$ satisfy the Eqs. 4 and 5, the server will calculate a cluster-wise cosine similarity matrix $\alpha_k$, and its entries are used as weights for building a full-connected graph with nodes being all clients within the cluster. The Stoer–Wagner minimum cut algorithm [35] is then applied to the constructed graph, which bi-partitions the graph and divides the cluster $\mathbb{C}_k \to \{\mathbb{C}_{k1}, \mathbb{C}_{k2}\}$. The clustering mechanism based on Eqs. 4 and 5 can automatically and dynamically determine the number of clusters along the FL, while the two hyper-parameters $\varepsilon_1$ and $\varepsilon_2$ can be easily set through some simple experiments on the validation sets following [33].

For a client $\mathbb{S}_i$ in cluster $\mathbb{C}_k$, it tries to find $\hat{\theta}_{k,i}$ that is close to the real solution $\theta_{k,i}^* = \arg\min_{\theta_i \in \Theta_k} f(\theta_{k,i}; \mathcal{G}_i)$. At a communication round $t$, the client $\mathbb{S}_k$ transmits its gradient to the server

$$\Delta\theta_{k,i}^t = \hat{\theta}_{k,i}^t - \theta_{k,i}^{t-1}. \tag{6}$$

Since the server maintains the cluster assignments, it can aggregate the gradients cluster-wise by

$$\theta_k^{t+1} = \theta_k^t + \sum_{i \in [n_k]} \Delta\theta_{k,i}^t. \tag{7}$$

### 4.4 Theoretical analysis

We investigate the problem of graph FL with multi-domain data distribution, and use the gradient-based FL paradigm [27] to facilitate the model training. We theoretically analyze that the gradient-based FL algorithm on GNNs can in principle reduce the structure and feature heterogeneity in clusters, along with the task difference between data from different domains. We study two general problems in order to prove that the gradients can reflect the feature, structure, and task information.

**Definition 4.1** *Let a function $f : \mathcal{X} \to \mathcal{Y}$ which maps from the metric space $(\mathcal{X}, d)$ to $(\mathcal{Y}, d')$, the function $f$ is considered to have $\delta$ distortion if $\forall u, v \in \mathcal{X}, \frac{1}{\delta} d(u, v) \leq d'(f(u), f(v)) \leq d(u, v)$.*

**Theorem 4.1** *(Bourgain theorem [1]) Given an $n$-point metric space $(\mathcal{X}, d)$ and an embedding function $f$ as defined above, $\forall u, v \in \mathcal{X}$, there exist an embedding mapped from $(\mathcal{X}, d)$ to $\mathbb{R}^k$ with the distortion of the embedding being $O(\log n)$.*

**Problem 1.** *In GCFL which involves the communication of the gradients between graphs with heterogeneous structures distributed among different clients, the structure and feature difference can be captured by the GNN gradients.*

For simplicity, we solve Problem 1 with the GNN of Simple Graph Convolutions (SGC) [41], through the following two propositions.

**Proposition 4.1** *Given a graph $G$ with fixed structure represented by the normalized graph Laplacian $\mathcal{L} = \widetilde{D}^{-\frac{1}{2}} \widetilde{A} \widetilde{D}^{-\frac{1}{2}}$, feature represented with $X$, and an SGC $f(\mathcal{L}, X) = softmax(\mathcal{L}^K X \Theta)$ with weights $\Theta$ trained on graph $G$. If we have another graph $G'$ with different structure $\mathcal{L}'$, the weight difference $||\Theta' - \Theta||_2$ is bounded with the structure difference.*

**Proposition 4.2** *Given a graph $G$ with fixed structure represented by the normalized graph Laplacian $\mathcal{L} = \widetilde{D}^{-\frac{1}{2}} \widetilde{A} \widetilde{D}^{-\frac{1}{2}}$, feature represented with $X$, and an SGC $f(\mathcal{L}, X) = softmax(\mathcal{L}^K X \Theta)$ with weights $\Theta$ trained on graph $G$. If we have another graph $G'$ with different feature $\mathcal{X}'$, the weight difference $||\Theta' - \Theta||_2$ is bounded with the feature difference.*

We prove proposition 4.1 and 4.2 in Appendix B. We use the *Bourgain theorem* to bound the difference between embeddings generated with different graph structures/features, and prove that the feature and structure information of a graph is incorporated into the model weights (gradients). By proving that the model weights (gradients) are bounded with the structure/feature difference, we show that the gradients will change with the structure and feature. This further justifies that our proposed gradient based clustering framework GCFL is able to capture the structure and feature information. In addition, we also study the following problem, which allows our GCFL framework to be further extended to cross-task graph-level federated learning in the future.

**Problem 2.** *The communicated gradients in GCFL can also capture the task heterogeneity.*

**Proposition 4.3** *Given a graph $G$ with structure represented by the normalized graph Laplacian $\mathcal{L} = \widetilde{D}^{-\frac{1}{2}} \widetilde{A} \widetilde{D}^{-\frac{1}{2}}$, and feature represented with $X$, if trained with different tasks, we will get the Simple SGC with bounded weights.*

The proof of proposition 4.3 can be found in Appendix B.

## 5  GCFL+: improved GCFL based on observation sequences of gradients

### 5.1  Fluctuation of gradient norms

When observing the norm of gradients for each communication round in GCFL, as shown in Figure 1, we notice that: 1) the norm of gradients continuously fluctuates; 2) different clients can have divergent

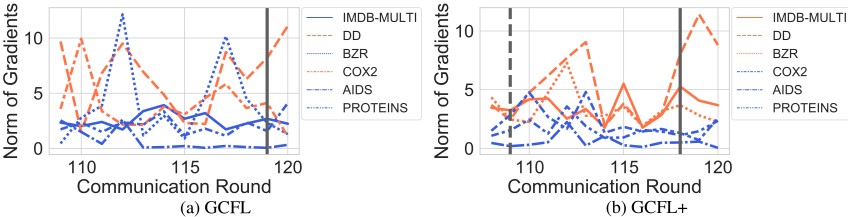

Figure 1: Norm of gradients versus communication round with six clients across datasets. Clients with datasets colored the same are split to the same cluster.

scales of gradient norms. The fluctuation of gradient norms and different scales indicate that the updating directions and distances of gradients for clients are diverse, which manifests the structure and feature heterogeneity in our setting again. In our vanilla GCFL framework, the server calculates a cosine similarity matrix based on the last transmitted gradients once the clustering criteria are satisfied. However, with the observation that the norm of gradients fluctuates along the communication round, albeit with the constraints of clustering criteria, GCFL clustering based on gradient-point could omit important client behaviors and be misled by noises. For example, in Figure 1 (a), GCFL performs clustering at round 119 based on the gradients at that round, which does not effectively find graphs with lower heterogeneity.

## 5.2 Technical design

Motivated by these observations, we propose an improved version of GCFL, named GCFL+, which conducts clustering by taking series of gradient norms into consideration. In the GCFL+ framework, the server maintains a multi-variant time-series matrix $Q \in \mathbb{R}^{\{n,d\}}$, where $n$ is the number of clients and $d$ is the length of a gradient series being tracked. At each communication round $t$, the server updates $Q$ by adding in the norm of gradients $\|\Delta\theta_i^t\|$ to $Q(i,:) \in \mathbb{R}^d$ and remove the out-of-date one. GCFL+ uses the same clustering criteria as GCFL (Eqs. 4 and 5). If the clustering criteria are satisfied, the server will calculate a distance matrix $\beta$ in which each cell is the pair-wise distance of two series of gradients. Here, we use a technique called dynamic time warping (DTW) [31] to measure the similarity between two data sequences. For a cluster $\mathbb{C}_k$, the server calculates its distance matrix as

$$\beta_k(p,q) = dist(Q(p,:), Q(q,:)), p, q \in idx(\{\mathbb{S}_i\}),\tag{8}$$

where $idx(\{\mathbb{S}_i\})$ is the indices of all clients $\{\mathbb{S}_i\}$ in cluster $\mathbb{C}_k$. With the distance matrix $\beta$, the server can perform bi-partitioning for clusters who meet the clustering criteria. As a result, in Figure 1 (b), GCFL+ performs clustering at round 118 based on the gradient sequence of length 10, which captures the longer-range behaviors of clients and effectively more homogeneous clusters.

# 6 Experiments

## 6.1 Experimental settings

**Datasets** We use a total of 13 graph classification datasets [30] from three domains including seven molecule datasets (MUTAG, BZR, COX2, DHFR, PTC_MR, AIDS, NCI1), three protein datasets (ENZYMES, DD, PROTEINS), and three social network datasets (COLLAB, IMDB-BINARY, IMDB-MULTI), each with a set of graphs. Node features are available in some datasets, and graph labels are either binary or multi-class. Details of the datasets are presented in Appendix C.

We design two settings that follow different data partitioning mechanisms, and the example real scenarios of the two settings can be found in Appendix A. The first setting (i.e., single-dataset) is to randomly distribute graphs from a single dataset to a number of clients, with each client holding a distinct set of about 100 graphs, among which 10% are held out for testing. In the second setting (i.e., multi-dataset), we use multiple datasets either from a single domain or multiple domains. Each client holds a graph dataset, among which 10% are held out for testing. In the first setting, we use NCI1, PROTEINS, and IMDB-BINARY from three domains and distribute them to 30, 10, 10 clients, respectively. In the second setting, we create three data groups including MOLECULES which consists of seven datasets from the molecule domain distributed into seven clients, BIOCHEM where we add three datasets from the protein domain into MOLECULES and distribute them into 10 clients, MIX where we add three datasets from the social domain into BIOCHEM and distribute them into 13 clients.

**Baselines** We use `self-train`[2] as the first baseline to test whether FL can bring improvements to each client through collaborative training. In `self-train`, each client firstly downloads the same randomly initialized model from the server and then trains locally without any communications. Then we implement two widely used FL baselines `FedAvg` [27] and `FedProx` [25], the latter of which can deal with data and system heterogeneity in non-graph FL. For the graph classification model, we use the same GIN [43] design, which represents the state-of-the-art GNN for graph-level tasks. We fix the GIN architecture and hyper-parameters through all baselines in order to control the experiments across different settings.

**Parameter settings** We use the three-layer GINs with hidden size of $64$. We use a batch size of $128$, and an Adam [20] optimizer with learning rate $0.001$ and weight decay $5e^{-4}$. The $\mu$ for `FedProx` is set to $0.01$. For all FL methods, the local epoch $E$ is set to $1$. The two important hyper-parameters $\varepsilon_1$ and $\varepsilon_2$ as clustering criteria vary in different groups of data, which are set through offline training for about $50$ rounds following [33]. We run all experiments for five random repetitions on a server with 8 24GB NVIDIA TITAN RTX GPUs.

## 6.2 Experimental results

**Federated graph classification within single datasets** Conceptually, clients in this setting are more homogeneous. As can be seen from the results in Table 3, our framework can obviously improve the performance of graph classification over local clients. For the NCI1 dataset distributed on 30 clients, GCFL and GCFL+ achieve $13.27\%$ and $14.75\%$ performance gains over `self-train` on average, and GCFL and GCFL+ help 10-14 more clients than `FedAvg` and `FedProx` who fail to improve about half of clients. For the PROTEINS dataset on the total 10 clients, GCFL and GCFL+ achieve $7.29\%$ and $7.81\%$ average performance gains compared to `self-train`. For IMDB-BINARY on 10 clients, `FedAvg` and `FedProx` fail to help 5/10 and 4/10 clients respectively, while both GCFL and GCFL+ are able to improve all 10 clients. Overall, `FedAvg` can only help around half of the clients, which demonstrates that `FedAvg` can be ineffective even for decenrtalized graphs from a single dataset, because of the graph non-IIDness as shown in Table 2. In addition, in all three datasets, the minimum performance gain of clients over `self-train` using GCFL or GCFL+ is obviously larger than the minimum performance gain using `FedAvg` and `FedProx`. It indicates that even when some clients do not improve from `self-train` by using GCFL or GCFL+, they can achieve more comparable performance as `self-train` than using `FedAvg` and `FedProx`. These experimental results demonstrate that our frameworks are effective on the single-dataset multi-client FL setting.

**Federated graph classification across multiple datasets** According to our data analysis in Tables 1 and 2, clients in such a setting are more heterogeneous. We conduct experiments with multiple datasets in two settings: single domain (using the data group MOLECULES), and across domains (using the data groups BIOCHEM and MIX). As can be seen from the results in Table 4, our frameworks GCFL and GCFL+ can effectively improve the performance of clients with distinct datasets. The results show $1.7\% - 2.7\%$ improvements of our frameworks compared to `self-train`. In all three data groups, our GCFL+ framework can improve twice as many as clients than `FedAvg`, and it achieves a ratio of $100\%$ in MOLECULES to improve all clients' performance. The `FedAvg` failed to improve around $60\%$ clients, which further demonstrates its ineffectiveness facing graph non-IIDness. Additionally, the GCFL+ framework also outperforms GCFL. In MIX, although GCFL can achieve the same ratio of improved clients as GCFL+, the GCFL+ framework has a much larger minimum gain of clients than GCFL. It indicates that by GCFL+ few clients that cannot benefit from others will not be degraded through the collaborating. These results indicate that graphs across datasets or even across domains are able to help each other through proper FL, which is a surprising and interesting start point for further study.

**Effects of hyper-parameters $\varepsilon_1$ and $\varepsilon_2$** The hyper-parameter $\varepsilon_1$ is a stopping criterion for checking whether a general FL on the current set of clients is near the stationary point. Theoretically, $\varepsilon_1$ should be set as small as possible. The hyper-parameter $\varepsilon_2$ is more dependent on the number of clients and the heterogeneity among them. A smaller $\varepsilon_2$ will make the clients more likely to be clustered. When $\varepsilon_1$ and $\varepsilon_2$ are in the feasible ranges, their small variation can have little effect on the performance because the clustering results would largely remain the same. When $\varepsilon_2$ is set too large,

---

[2] We do not have a global model as baseline as datasets are from various domains and their tasks are divergent.

Table 3: Performance on the single-dataset-multi-client setting. We present the average accuracy and minimum gain over `self-train` on all clients, as well as the ratio of clients which get improved.

| Dataset (# clients) | NCI1 (30) | | | PROTEINS (10) | | | IMDB-BINARY (10) | | |
|---|---|---|---|---|---|---|---|---|---|
| Accuracy | average | min gain | ratio | average | min gain | ratio | average | min gain | ratio |
| `self-train` | 0.6468($\pm$0.053) | — | — | 0.7213($\pm$0.058) | — | — | 0.7654($\pm$0.057) | — | — |
| FedAvg | 0.6474($\pm$0.076) | -0.1333 | 14/30 | 0.7490($\pm$0.034) | -0.0615 | 6/10 | 0.7596($\pm$0.049) | -0.0800 | 5/10 |
| FedProx | 0.6437($\pm$0.072) | -0.2400 | 16/30 | 0.7556($\pm$0.036) | -0.0923 | 7/10 | 0.7746($\pm$0.048) | -0.0600 | 6/10 |
| GCFL | 0.7326($\pm$0.052) | -0.0462 | 26/30 | 0.7739($\pm$0.043) | -0.0545 | 8/10 | 0.8256($\pm$0.059) | 0.0182 | **10**/10 |
| GCFL+ | **0.7422**($\pm$0.053) | -0.1143 | **28**/30 | **0.7776**($\pm$0.037) | -0.0154 | **9**/10 | **0.8299**($\pm$0.052) | 0.0167 | **10**/10 |

Table 4: Performance on the multi-dataset-multi-client setting. Metrics are the same as Table 3.

| Dataset (# domains) | MOLECULES (1) | | | BIOCHEM (2) | | | MIX (3) | | |
|---|---|---|---|---|---|---|---|---|---|
| Accuracy | average | min gain | ratio | average | min gain | ratio | average | min gain | ratio |
| `self-train` | 0.7543($\pm$0.017) | — | — | 0.7129($\pm$0.016) | — | — | 0.7001($\pm$0.034) | — | — |
| FedAvg | 0.7524($\pm$0.026) | -0.0132 | 3/7 | 0.6944($\pm$0.027) | -0.1467 | 4/10 | 0.6886($\pm$0.023) | -0.1233 | 5/13 |
| FedProx | 0.7668($\pm$0.032) | -0.0054 | 5/7 | 0.7053($\pm$0.026) | -0.1000 | 5/10 | 0.6897($\pm$0.026) | -0.1367 | 5/13 |
| GCFL | 0.7661($\pm$0.016) | 0.0010 | **7**/7 | 0.7172($\pm$0.019) | -0.0700 | 7/10 | 0.7056($\pm$0.019) | -0.1400 | **10**/13 |
| GCFL+ | **0.7745**($\pm$0.030) | 0.0010 | **7**/7 | **0.7312**($\pm$0.031) | -0.0300 | **8**/10 | **0.7121**($\pm$0.021) | -0.0233 | **10**/13 |

the performance will be similar as applying a basic FL algorithm directly (i.e. with a single cluster). When $\varepsilon_2$ is set too small, more clusters with smaller sizes or even single clients will be generated. We provide additional experimental results regarding the performance of GCFL and GCFL+ w.r.t. varying $\varepsilon_1$ and $\varepsilon_2$ in Figure 2. As shown in the subfigures, some points that represent varying $\varepsilon_2$ overlap for each $\varepsilon_1$, which indicates that varying $\varepsilon_2$ in a certain range w.r.t. the fixed $\varepsilon_1$ leads to similar performance. Looking at a $\varepsilon_2$, within a certain range of $\varepsilon_1$, we can find the performance often fluctuating within a 0.01 variance. The results show that the performance of GCFL and GCFL+ are not very sensitive to the changes of $\varepsilon_1$ and $\varepsilon_2$ in reasonable ranges.

### 6.3 Structure and feature analysis in clusters

We conduct an in-depth analysis to explore the clustering results of GCFL and GCFL+. As can be seen in Figure 3, after being clustered by GCFL and GCFL+, the overall structure and feature heterogeneity of clients' graphs within clusters are reduced significantly compared to the original values, especially for the multiple dataset setting (Figure 3c and 3d). For the one dataset setting (Figure 3a and 3b), since features all fall in the same space, pairs of clients tend to have more homogeneous features. Therefore, the feature heterogeneity only gets reduced slightly after clustering. Unlike feature heterogeneity, the structure heterogeneity within clusters decreases significantly. In the setting of multiple datasets, as shown in Figure 3c and 3d, both structure and feature heterogeneity decrease significantly, which is intuitive since datasets across domains usually tend to have higher heterogeneity, as discussed in 4.1. We also look into the clusters and find that datasets from the same domains are more likely to be clustered together, while datasets from different domains also constantly get clustered together and benefit each other. For example, the clustering of GCFL+ corresponding to Figure 3d groups two social networks COLLAB and IMDB-BINARY together with PROTEINS and also several molecules datasets, and there is also a cluster of NCI1, DD, and IMDB-MULTI which are molecules, proteins and social networks, respectively. These analysis manifests that domains of datasets can verify the sanity of clusters to some extent, but one cannot solely rely on such prior knowledge to determine the optimal clusters, which demonstrates the necessity of our frameworks with the ability of performance-driven dynamic clustering along the process of FL.

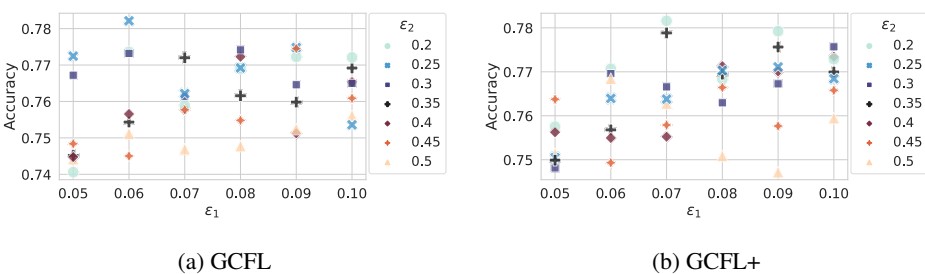

|(a) GCFL|(b) GCFL+|

Figure 2: Performance of GCFL and GCFL+ on MOLECULES w.r.t varying $\varepsilon_1$ and $\varepsilon_2$.

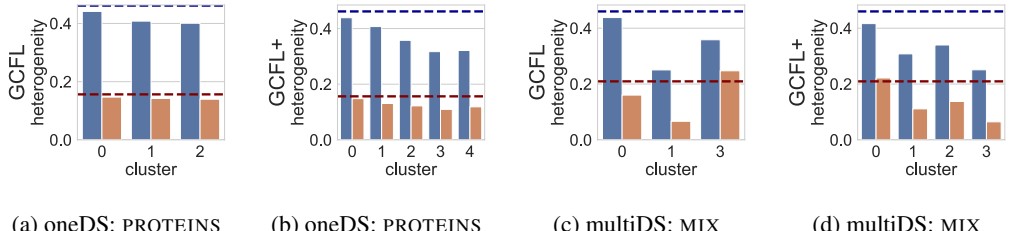

| (a) oneDS: PROTEINS | (b) oneDS: PROTEINS | (c) multiDS: MIX | (d) multiDS: MIX |

Figure 3: Structure (blue) and feature (red) heterogeneity within clusters found by GCFL and GCFL+. Dashed lines denote the heterogeneity over all clients before clustering.

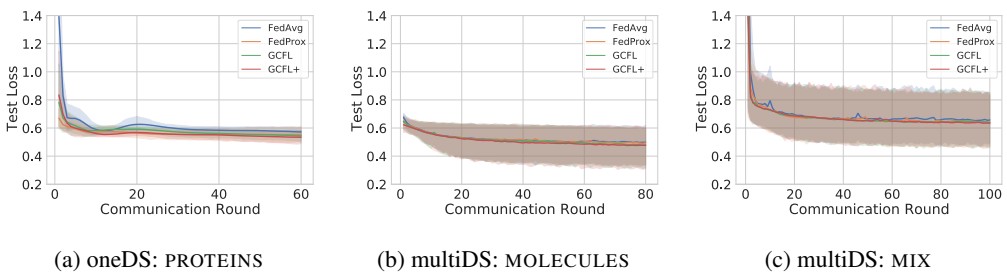

| (a) oneDS: PROTEINS | (b) multiDS: MOLECULES | (c) multiDS: MIX |

Figure 4: Average with standard deviation of the training curves of all clients.

## 6.4 Convergence analysis

We visualize the testing loss with respect to the communication round to show the convergence of GCFL and GCFL+ compared with the standard federated learning baselines. Figure 4 shows the training curves on two settings, which illustrates that GCFL and GCFL+ achieves similar convergence rate as FedProx, which is the state-of-the-art FL framework dealing with non-IID Euclidean data. We also notice that both GCFL, GCFL+ and FedProx can converge to a lower loss compared with FedAvg, which corroborates our consideration of the non-IID problem in our setting.

## 6.5 More results in Appendix

In Table 3 and 4, we averaged the accuracy across all clients for presentation simplicity. To understand the detailed performance by clients and clusters, we present different Violin plots in Appendix D. Besides, we also show more results regarding various settings (overlapping clients, real vs. synthetic node features, standardized gradient-sequence matrix in GCFL+, etc) in Appendix D.

## 7 Conclusion

In this work, we propose a novel setting of cross-dataset and cross-domain federated graph classification. The techniques (GCFL and GCFL+) we develop allow multiple data owners holding structure and feature non-IID graphs to collaboratively train powerful graph classification neural networks without the need of direct data sharing. As the first trial, we focus on the effectiveness of FL in this setting and have not carefully studied other issues such as data privacy, although it is intuitive to preserve the privacy of clients by introducing an encryption mechanism (e.g. applying orthonormal transformations), and to prevent from adversarial scenarios by clustering out the malicious clients. Due to its evident motivations and proofs on the effective FL in a new setting, we believe this work can serve as a stepping stone for many interesting future studies.

## Acknowledgments and Disclosure of Funding

The work is partially supported by National Science Foundation (NSF) under CNS-2124104, CNS-2125530, CNS-1952192, and IIS-1838200, National Institute of Health (NIH) under R01GM118609 and UL1TR002378, and the internal funding and GPU servers provided by the Computer Science Department of Emory University.

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
