# A More Motivating Examples and Analysis

**Additional explanations for Table 1**  We provide two real examples of how graph properties can decide important graph patterns as the signals for graph classifications. The first example is about social networks. Among the graph properties, social networks (such as IMDB-BINARY shown in Table 1) have a significant high clustering coefficient (CC), which means they contain many triangles as a graph pattern. This may be an important signal for effective graph classification, which is consistent with commonly used social network classification methods such as motif-counting. On the other hand, if a set of random graphs does not possess high CC, the prominent graph patterns might be some subtrees or long edges, where the training of a graph classification model is not likely to facilitate the capturing of triangles, and thus not beneficial for the classification of social networks. The second example is about protein networks and superpixel networks (such as ENZYMES and MSRC_21 shown in Table 1). ENZYMES is a protein dataset, in which a graph is an enzyme with secondary structure elements as nodes and the links connect nodes if they are neighbors in space. MSRC_21 is a semantic image processing dataset, in which nodes represent superpixels and links are constructed if two superpixels are adjacent. Although the two datasets are from totally different domains, they are both formed by spatial structures, which is reflected in Table 1 where the two datasets have very close values regarding multiple properties (degree distribution, shortest path length, CC). For ENZYME, it is known that the tertiary structure of proteins is essential and necessary for their biological activities, and the tightly knit groups are important signals for classifying an enzyme into different catalyzed levels. For MSRC_21, the environment information is also essential for identifying the superpixels or describing objects in the images. Thus, there exists the case that two cross-domain datasets contain certain significant graph patterns that correspond to different dataset-specific meanings and tasks, but can be shared across datasets towards the training of more powerful graph models.

**Additional analysis for clustered FL**  With analysis from Table 2, it shows significant statistical heterogeneity regarding features and structures in graphs. To motivate clustered FL, we conduct additional experiments regarding the connections between such statistical heterogeneity and the power of GNNs. Specifically, we use PROTEINS as one example dataset, and pair it with other datasets from the same and different domains. We focus on structure heterogeneity here and use one-hot degree features in order to avoid the effects of feature heterogeneity. We then train a GIN model with basic FedAvg on all pairs, and analyze the GIN performance versus the structural heterogeneity as defined in our paper on PROTEINS in Figure 5. As observed from Figure 5: (1) at first, the GNN on one graph dataset (PROTEINS) can benefit from the federated learning on another graph dataset, and that benefit becomes larger as the

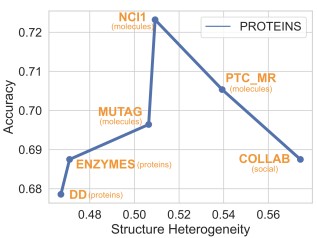

Figure 5: GIN Performance on PROTEINS (proteins) federated averagely trained with datasets from the same or different domains.

structure heterogeneity between two graphs becomes larger (DD, ENZYMES, MUTAG, NCI1), which clearly supports the benefit of cross-dataset/cross-domain graph federated learning. However, (2) as the heterogeneity becomes too large, the performance of GNN starts to degenerate (PTC_MR, COLLAB), which clearly supports our design of clustered federated learning. Our whole framework is built to achieve a good trade-off between (1) and (2), which we achieved to some extent, but can further improve on it in future studies.

**Example real scenario for the single-dataset setting**  In this work, we use the public IMDB datasets to mimic the real scenario for social networks. For example, TikTok as an emerging video sharing platform has branches in different countries nowadays, among which direct data sharing is illegal. However, the users in different countries naturally reside in social networks with similar properties. It is then very viable to perform graph federated learning across branches in different countries towards the training of more powerful graph learning models for tasks such as group/community profiling.

**Example real scenario for multi-dataset setting**  In this work, we synthesized the extreme setting where each client holds data from totally different domains for experimental purposes, to establish the benefit of cross-domain collaboration. A realistic scenario in a less extreme setting can be about various types of apps on mobile phones. For example, different types of iPhone users may leverage

healthcare apps and social media apps and opt in their data collection. As a consequence, the two apps will both hold some user data that may complement each other but cannot be directly shared across the departments. Since Apple owns both departments, it is then easier for them to collaborate through a federated learning framework.

## B  Missing Proofs in Section 4.4

### B.1  Proof of Proposition 4.1

Assume the structure difference between graph $G$ and $G'$ is bounded with

$$||\mathcal{L}' - \mathcal{L}||_2^2 = ||E_\mathcal{L}||_2^2 \leq \epsilon_\mathcal{L}, \tag{9}$$

The difference between the original weights $\Theta$ and the weights trained on the new graph structure $\Theta'$ is represented as

$$
\begin{aligned}
||\Theta' - \Theta||_2^2 &= ||(\mathcal{L}'X)^{-1}Y' - (\mathcal{L}X)^{-1}Y||_2^2 \\
&= ||X^{-1}(\mathcal{L}'^{-1}Y' - \mathcal{L}^{-1}Y)||_2^2.
\end{aligned} \tag{10}
$$

Given that $||\mathcal{L} \cdot \mathcal{L}'||_2^2 = ||\mathcal{L} \cdot (\mathcal{L} + E_\mathcal{L})||_2^2 \geq ||\mathcal{L}E_\mathcal{L}||_2^2$. Let $||\mathcal{L}E_\mathcal{L}||_2^2 = \delta_\mathcal{L}$, then we can get $||\mathcal{L}'^{-1} - \mathcal{L}^{-1}||_2^2 = ||E_{\mathcal{L}^{-1}}|| \leq \frac{\epsilon_\mathcal{L}}{\delta_\mathcal{L}}$.

Given the Bourgain theorem [50], the difference between the embedding $Y$ and $Y'$ is bounded with

$$||\mathcal{Y}' - \mathcal{Y}||_2^2 = ||E_Y||_2^2 \leq \epsilon_Y, \tag{11}$$

The weight difference can then be bounded with

$$
\begin{aligned}
||\Theta' - \Theta||_2^2 &\leq ||X^{-1}||_2^2 ||(\mathcal{L}^{-1} + E_{\mathcal{L}^{-1}})(Y + E_Y) - \mathcal{L}^{-1}Y||_2^2 \\
&= ||X^{-1}||_2^2 ||\mathcal{L}^{-1}E_Y + E_{\mathcal{L}^{-1}}Y + E_{\mathcal{L}^{-1}}E_Y||_2^2 \\
&\leq ||X^{-1}||_2^2 \left[ \epsilon_Y ||\mathcal{L}^{-1}||_2^2 + \frac{\epsilon_\mathcal{L}}{\delta_\mathcal{L}} ||Y||_2^2 + \frac{\epsilon_\mathcal{L}\epsilon_Y}{\delta_\mathcal{L}} \right].
\end{aligned} \tag{12}
$$

With the trained SGC, the feature and graph structure is fixed as $X$ and $\mathcal{L}$. Thus the weight difference is bounded with $X$ and $\mathcal{L}$.

### B.2  Proof of Proposition 4.2

Assume the feature difference between graph $G$ and $G'$ is bounded with

$$||X' - X||_2^2 = ||E_X||_2^2 \leq \epsilon_X, \tag{13}$$

The difference between the original weights $\Theta$ and the weights trained on the new graph structure $\Theta'$ is represented as

$$||\Theta' - \Theta||_2^2 = ||(\mathcal{L}X')^{-1}Y' - (\mathcal{L}X)^{-1}Y||_2^2 \tag{14}$$

Given that $||X \cdot X'||_2^2 = ||X \cdot (X + E_X)||_2^2 \geq ||XE_X||_2^2$. Let $||XE_X||_2^2 = \delta_X$, then we can get $||X'^{-1} - X^{-1}||_2^2 = ||E_{X^{-1}}|| \leq \frac{\epsilon_X}{\delta_X}$.

Given the Bourgain theorem [50], the difference between the embedding $Y$ and $Y'$ is bounded with

$$||\mathcal{Y}' - \mathcal{Y}||_2^2 = ||E_Y||_2^2 \leq \epsilon_Y, \tag{15}$$

The weight difference can then be bounded with

$$
\begin{aligned}
||\Theta' - \Theta||_2^2 &= ||(X'^{-1}\mathcal{L}^{-1}Y' - X^{-1}\mathcal{L}^{-1}Y||_2^2 \\
&= ||(X'^{-1}\mathcal{L}^{-1}(Y + E_Y) - X^{-1}\mathcal{L}^{-1}Y||_2^2 \\
&= ||(X'^{-1}\mathcal{L}^{-1}Y + X'^{-1}\mathcal{L}^{-1}E_Y - X^{-1}\mathcal{L}^{-1}Y||_2^2 \\
&= ||(X'^{-1}\mathcal{L}^{-1}Y - X^{-1}\mathcal{L}^{-1}Y + X'^{-1}\mathcal{L}^{-1}E_Y||_2^2 \\
&= ||(X'^{-1} - X^{-1})\mathcal{L}^{-1}Y + X'^{-1}\mathcal{L}^{-1}E_Y||_2^2 \\
&= ||E_{X^{-1}}\mathcal{L}^{-1}Y + (X + E_X)^{-1}\mathcal{L}^{-1}E_Y||_2^2 \\
&= ||E_{X^{-1}}\mathcal{L}^{-1}Y + (\mathcal{L}X + \mathcal{L}E_X)^{-1}E_Y||_2^2 \\
&\leq \frac{\epsilon_X}{\delta_X}||\mathcal{L}^{-1}Y||_2^2 + \frac{\epsilon_X^2\epsilon_Y}{\delta_X}||(\mathcal{L}X)^{-1}||_2^2 + \epsilon_X\epsilon_Y||(\mathcal{L}X)^{-1}||_2^4
\end{aligned} \tag{16}
$$

With the trained SGC, the feature and graph structure is fixed as $X$ and $\mathcal{L}$. Thus the weight difference is bounded.

### B.3 Proof of Proposition 4.3

The difference between the weights trained with different tasks can be written as

$$||\Theta_i - \Theta_j||_2^2 = ||(\mathcal{L}X)^{-1}(Y_i - Y_j)||_2^2. \tag{17}$$

With the Bourgain theorem, we have the transformed embedding bounded with

$$||Y_i - Y_j||_2^2 = ||E_Y||_2^2 \le \epsilon_Y. \tag{18}$$

By substituting $||Y_i - Y_j||_2^2 = ||E_Y||_2^2$, we have

$$\begin{aligned}||\Theta_i - \Theta_j||_2^2 &= ||(\mathcal{L}X)^{-1}E_Y||_2^2 \\ &\le \epsilon_Y ||(\mathcal{L}X)^{-1}||_2^2 \end{aligned} \tag{19}$$

## C  Dataset Details

We provide details of the datasets we use in Table 5.

Table 5: The statistics of datasets.

| dataset | statistics | | | | | dataset | statistics | | | | |
|---------|---------|-------------|-------------|----------|---------------|---------|---------|-------------|-------------|----------|---------------|
| | #graphs | avg. #nodes | avg. #edges | #classes | node features | | #graphs | avg. #nodes | avg. #edges | #classes | node features |
| MUTAG | 188 | 17.93 | 19.79 | 2 | original | ENZYMES | 600 | 32.63 | 62.14 | 6 | original |
| BZR | 405 | 35.75 | 38.36 | 2 | original | DD | 1178 | 284.32 | 715.66 | 2 | original |
| COX2 | 467 | 41.22 | 43.45 | 2 | original | PROTEINS | 1113 | 39.06 | 72.82 | 2 | original |
| DHFR | 467 | 42.43 | 44.54 | 2 | original | COLLAB | 5000 | 74.49 | 2457.78 | 3 | degree |
| PTC_MR | 344 | 14.29 | 14.69 | 2 | original | IMDB-BINARY | 1000 | 19.77 | 96.53 | 2 | degree |
| AIDS | 2000 | 15.69 | 16.20 | 2 | original | IMDB-MULTI | 1500 | 13.00 | 65.94 | 3 | degree |
| NCI1 | 4110 | 29.87 | 32.30 | 2 | original | | | | | | |

## D  More Detailed Experiment Results

**Violin plots instead of tables**  Figures 6, 7, and 8 show the detailed experiment results regarding more various client settings including overlapped vs. non-overlapped data partitioning, original vs. synthetic node features, standardized vs. non-standardized multi-variant time-series matrix in GCFL+. Since we want to provide more detailed results by clients, we use violin plots to display the distributions of performance gains of all clients compared to `self-train`, instead of the average numbers in Tables 3 and 4. In Figure 6 and 7, each violin represents a distribution of all clients' performance gain using one algorithm, and in Figure 8 each violin represents a distribution of all clients' performance gain on one dataset or data group. In Figures 6, 7, and 8, the blue left sides of violins are corresponding to the results in the main tables 3 and 4.

**Overlapping versus non-overlapping**  For distributing one dataset to multiple clients, we compare the two settings of allowing overlapping (same graphs appearing multiple clients) and not. As can be seen in Figure 6, our frameworks can also improve on overlapped clients.

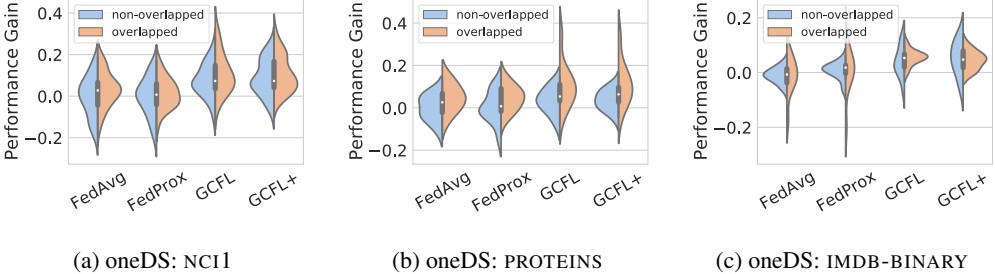

(a) oneDS: NCI1     (b) oneDS: PROTEINS     (c) oneDS: IMDB-BINARY

Figure 6: Distributions of performance gains of all clients with overlapped versus non-overlapped data partitioning.

**Original node features versus one-hot degree** features Apart from the original node features, we also use one-hot node degree features, in order to study the influence of node features. Figure (7a, 7b) and (7c, 7d) show the comparisons between original features and one-hot degree features on the single-dataset (oneDS) setting and the multi-dataset (multiDS) setting, respectively. Overall, our frameworks can consistently improve when using one-hot degree features. However, as in Figure 7c, the performance gains decreased when using only one-hot node degrees, which can be because of the decease of feature heterogeneity.

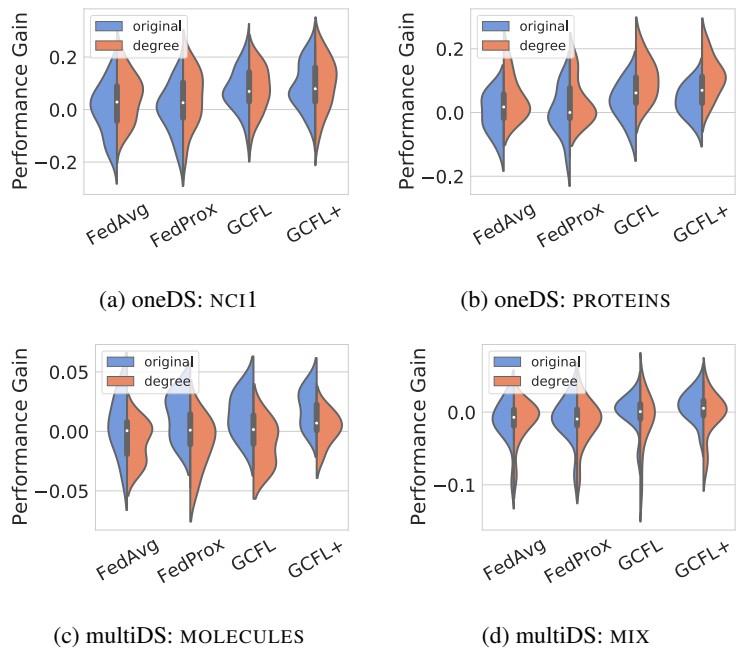

(a) oneDS: NCI1

(b) oneDS: PROTEINS

(c) multiDS: MOLECULES

(d) multiDS: MIX

Figure 7: Distributions of performance gains of all clients using original node features versus one-hot degree features on the oneDS (top) and multiDS (bottom) settings.

**Standardized versus non-standardized multi-variant time-series matrix** For the GCFL+ framework, we compared the performance gains of clients using the standardized or non-standardized multi-variant time-series matrix $Q \in \mathbb{R}^{\{n,d\}}$. By standardization, only the trends of gradients' fluctuation are considered and the scales are ignored. The standardization step is performed before calculating the distance matrix $\beta$ as

$$Q'(i,:) = Q(i,:)/std(Q(i,:)), i = 0, 1, \ldots, n. \tag{20}$$

As shown in Figure 8, the average performance gains of standardization and non-standardization are similar.

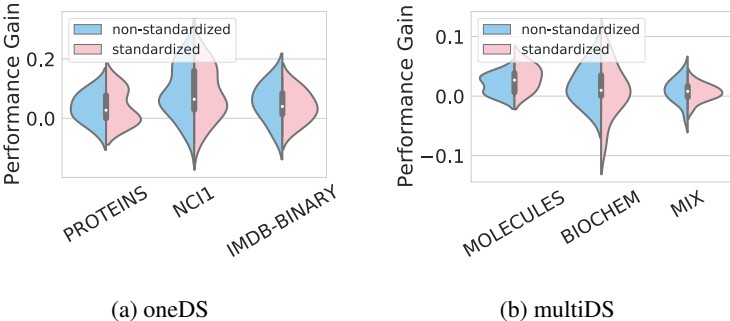

(a) oneDS

(b) multiDS

Figure 8: Distributions of performance gains of all clients using standardized gradient-sequence matrix versus non-standardized gradient-sequence matrix in GCFL+ on the oneDS and multiDS settings.