# OpenReview forum: "Federated Graph Classification over Non-IID Graphs"
_NeurIPS.cc/2021/Conference — NeurIPS 2021 Poster_

### Official Review · Reviewer_S61D · 2021-07-13

**Rating:** 4
**Confidence:** 3

**Summary:**

The paper proposes a federated graph classification system based on clustered federated learning to solve the non-iid problem in graph structure and node features across clients. The paper also proposes to use dynamic time warping to improve clustering performance. The authors evaluate their system on a couple of different scenarios: (i) a single graph dataset split across multiple clients and (ii) Graph datasets from different domains on each client.


**Limitations And Societal Impact:**

No comments here to adress.

**Main Review:**

Summary

The paper proposes a federated graph classification system based on clustered federated learning to solve the non-iid problem in graph structure and node features across clients. The paper also proposes to use dynamic time warping to improve clustering performance. The authors evaluate their system on a couple of different scenarios: (i) a single graph dataset split across multiple clients and (ii) Graph datasets from different domains on each client.

Pros

The authors recognize an important problem in the non-iidness that occurs in graph datasets within the same domain and naturally across domains. The paper is mostly well written and easy to follow, although a few sections could benefit from clear phrasing.

Cons

The main concern with the paper are a lack of technical novelty, an unconvincing motivation, and inadequate experimental evaluation. Detailed comments are given below.


Detailed comments:
It's unclear what the comparison of random graphs to real world graphs is meant to illustrate beyond the fact that real world graphs illustrate certain common properties across domains. This is a fairly well understood phenomenon and it seemed there was not much insight with this data in the introduction.

The motivation for clustering is entirely dependent on a robust metric of homogeneity. On the one hand, statistical structural properties (as shown in Table 1) might be similar between graphs on different clients. But how much do these statistical properties matter? What is the correlation between the expressivity of a GNN trained on different graphs and the statistical similarity between those graphs?


GCFL
"From Table 1 we notice that real-world graphs tend to share certain general properties across different graphs, datasets and even domains..". This is probably a bit strong claim to be made from the Table 1. The information presented in Table 1 is inadequate to make this claim. At the very least more datasets need to be investigated.  Table 1 in itself doesn't seem like a strong enough motivation for clustering.

 Graph Clustered Federated Learning needs to be motivated with a real world use case that satisfies these properties. Where do real world scenarios with different clients containing heterogeneous features in their graphs / nodes show up? Why would an IMDB-BINARY type dataset ever be split between clients that cannot share data with each other? Or when would a client working on molecular data ever collaborate with a social media company working on social networks? Recent work https://arxiv.org/abs/2104.07145 motivated federated graph learning in the context of molecular graphs, but the features that are used in such a setting are often atom properties which would be fairly homogenous between graphs and clients. So the motivation needs to be better.

The technical novelty is limited. Section 4.3 seems more like an application of CFL than a novel approach to clustered federated learning. Moreover, the approach seems general and doesn't seem specific to exploiting any GNN related specifics.

GCFL+
Observations on gradient norm and why GCFL cannot adjust itself makes sense, still a general phenomenon in FL.
Using DTW is simple and neat, but experimentally, improvement seems not significant. Have you ever tried different window lengths ?

Experiments
In the single dataset to number of clients setting, how is the dataset partitioned in a non-iid fashion between clients?

The baselines are not strong enough. What is the motivation for the self-train baseline?

How does GCFL compare to just CFL being applied?

The claim that GCFL can help clients that are training on disparate graphs (different domains) collaborate needs to be better justified. Maybe I misunderstood, but it seems that  the authors are suggesting that the semantic information in the graphs on each client is  irrelevant? I am not sure if a client training on social network data can collaborate with a client training on some other  GCN models. Hence, collaboration across domains may not be that trivial, in my view. The experimentsalsection needs to be reinforced with  more combinations of datasets from different domains to convince readers  that such a non-intuitive result holds true in practice.

Proofs
Why is the model used for analysis SGC but the model used in experiments GIN? I understand that it is for the sake of simplicity, but the models are quite different. Do the proofs hold up if the model being analyzed was GIN? Some intuition on this, even if not a full proof would definitely clear things up.


**Time Spent Reviewing:**

2

---

> ### Author Response · Authors · 2021-08-10
> **Initial response to Reviewer S61D**
>
> We thank the reviewer for the detailed comments on our paper, which acknowledge the importance of our problem and suggest several ways of improvement. Regarding the reviewer’s confusions and concerns, we provide the following clarifications and improvement plans.
>
> ## 1. Cross-domain graph federated learning is ***Beneficial***
> 1.1 Regarding the reviewer’s confusion about how Table 1 motivates cross-domain graph federated learning, we provide two real examples of how graph properties can decide important graph patterns as the signals for real graph classification tasks.
>
> The first example is about social networks. Among the graph properties, social networks (such as IMDB-BINARY shown in Table 1) have a significantly high clustering coefficient (CC), which means they contain many triangles as a graph pattern. This may be an important signal for effective graph classification, which is consistent with commonly used social network classification methods such as motif-counting. On the other hand, if a set of random graphs does not possess high CC, the prominent graph patterns might be some subtrees or long edges, where the training of a graph classification model is not likely to facilitate the capturing of triangles, and thus not beneficial for the classification of social networks.
>
> The second example is about protein networks and superpixel networks (such as ENZYMES and MSRC_21 shown in Table 1). ENZYMES is a protein dataset, in which a graph is an enzyme with secondary structure elements as nodes and the links connect nodes if they are neighbors in space. MSRC_21 is a semantic image processing dataset, in which nodes represent superpixels and links are constructed if two superpixels are adjacent. Although the two datasets are from totally different domains, they are both formed by spatial proximity, which is reflected in Table 1 where the two datasets have very close values regarding multiple properties (degree distribution, shortest path length, CC). For ENZYMES, it is known that the tertiary structure of proteins is essential and necessary for their biological activities, and the tightly knit groups are important signals for classifying an enzyme into different catalyzed levels. For MSRC_21, the environment information is also essential for identifying the superpixels or describing objects in the images. Thus, there exists the case that two cross-domain datasets contain certain significant graph patterns that correspond to different dataset-specific meanings and tasks, but can be shared across datasets towards the training of more powerful graph models.
>
> We will include discussions of such examples at least in the appendix in a revision.
>
> 1.2 The reviewer is also concerned about why clustering is needed in cross-domain graph federated learning. First of all, we are not motivated by Table 1 alone to perform clustering, but rather the additional analysis in Table 2, which shows significant statistical heterogeneity regarding features and structures in graphs, despite the shared graph properties shown in Table 1. Then, as suggested by the reviewer, we provide additional experimental results regarding the connections between such statistical heterogeneity among graphs and the power of GNNs trained on them. Specifically, we use PROTEINS as one dataset, and pair it with all other datasets from the same and different domains. We train a GIN model with basic FedAvg on all pairs of them, and record the structural heterogeneity as defined in our paper versus the GIN performance on PROTEINS as follows.
>
> | Pair dataset    | DD     | ENZYMES | IMDB-B | COX2   | IMDB-M | PTC_MR | COLLAB |
> |-----------------|--------|---------|--------|--------|--------|--------|--------|
> | Heterogeneity   | 0.4615 | 0.4664  | 0.5041 | 0.5100 | 0.5123 | 0.5405 | 0.5709 |
> | GIN performance | 0.6054 | 0.5946  | 0.6107 | 0.6446 | 0.6232 | 0.6071 | 0.6018 |
>
> As we can clearly observe, (1) at first, the GNN on one graph (PROTEINS) can benefit from the federated learning on another graph, and that benefit becomes larger as the heterogeneity between two graphs becomes larger (ENZYMES, IMDB-B, COX2), which clearly supports the benefit of cross-dataset/cross-domain graph federated learning. However, (2) as the heterogeneity becomes too large, the performance of GNN starts to degenerate (IMDB-M, PTC_MR, COLLAB), which clearly supports our design of clustered federated learning. Our whole framework is built to automatically achieve a good trade-off between (1) and (2), which we achieved to some extent, but can further improve on it in future studies. We thank the reviewer for pointing this out, and agree that it would be beneficial to include more comprehensive studies like these in a revision.
>
> 1.3 In our experiments, we have already included 13 datasets from three different domains, and the results have shown significant and consistent improvements of our proposed methods. Our baselines are indeed simple, but there are basically no stronger baselines due to the infancy of research in this direction. In fact, self-train is not a weak baseline at all, because our backbone model GIN is already state-of-the-art on single-dataset graph classification. Any significant and consistent improvement on top of GIN is not trivial, which exactly showcases the universal benefit of cross-domain graph federated learning and the effectiveness of our designed frameworks.
>
> ## 2. Cross-domain graph federated learning is ***Realistic***
> 2.1 The reviewer questioned why it is realistic to split a single graph dataset such as IMDB-BINARY among multiple clients. We used the public IMDB datasets in this work for experimental purposes. Here we give a more realistic example. TikTok as an emerging video sharing platform has branches in different countries nowadays, among which direct data sharing is illegal. However, the users in different countries naturally reside in social networks with similar properties. It is then very viable to perform graph federated learning across branches in different countries towards the training of more powerful graph learning models for tasks such as group/community profiling, especially when the labeled data in any branches alone are limited.
>
> 2.2 The reviewer also questioned why cross-domain collaboration is realistic. We used the extreme setting where each client holds data from totally different domains, again for experimental purposes, to establish the benefit of cross-domain collaboration. One step back, we give a realistic example in a less extreme setting. Different types of iPhone users may leverage healthcare apps and social media apps and opt in their data collection. As a consequence, the two apps will both hold some user data that may complement each other but cannot be directly shared across the departments. Since Apple owns both departments, it is then tempting for them to collaborate through a secure federated learning framework.
>
> ## 3. Cross-domain graph federated learning is ***Challenging***
> 3.1 First of all, as recognized by all reviewers, the establishment of cross-domain graph federated learning itself is new and non-trivial, which is done through tremendous data analysis and reasoning.
>
> 3.2 Technically speaking,
> * Research on federated learning itself is at its infancy. Although there are many emerging methods in other domains with non-graph data, classic general frameworks such as FedAvg and FedProx do not work well on graph data as we show in the experiments, while the more domain-specific frameworks do not trivially generalize to graph data.
> * Our GCFL framework, although in its essence is indeed a combination of CFL and GNN plus a novel modification, is concretely supported by thorough data analysis and rigorous theoretical analysis. Extensive experimental results also show clear improvements brought by our designs.
> * We are aware of many promising directions to further improve the technical designs. However, after all, the goal in this work was not to develop a fancy complicated method, but to establish a novel setting with the well-understood basic techniques, as a valid start point for follow-up research.
>
> ## 4. Other smaller issues
>
> 4.1 In the single dataset experiments, the graphs are partitioned randomly, so there is no controlled heterogeneity (some are more similar while some not not). We want to show the algorithm can properly cluster the clients based on their different heterogeneity.
>
> 4.2 Experiments with GCFL+. Although the absolute accuracy improvements are not always significant, GCFL+ is able to help more clients as can be seen from the min-gain and ratio columns in Tables 3&4. As we can observe from below, the performance of GCFL+ does not change much when the window size in DTW changes, as long as the windows with reasonable sizes are applied.
>
> | Window size | 5      | 10     | 20     |
> |-------------|--------|--------|--------|
> | NCI1        | 0.6267 | 0.6242 | 0.6220 |
> | PROTEINS    | 0.6752 | 0.6800 | 0.6736 |
> | IMDB-BINARY | 0.6407 | 0.6374 | 0.6401 |
>
> 4.3 SGC is used in our theoretical analysis because it provides a simplified approximation to GNNs with nonlinear layers which enables the subsequent rigorous analysis in Section 4.4. SGC has established comparable effects with nonlinear GNNs in many settings, and it is a common practice to use SGC to mimic the behaviors of GNNs in theoretical studies [2, 3].
>
> ## References
> [1] Caldarola, Debora, et al. “Cluster-driven graph federated learning over multiple domains.”  CVPR21 Workshop Learning from Limited or Imperfect Data (L^2ID). 2021.
> [2] Zhao, Lingxiao, and Leman Akoglu. "PairNorm: Tackling Oversmoothing in GNNs." ICLR. 2019.
> [3] Chang, Heng, et al. "A restricted black-box adversarial framework towards attacking graph embedding models." AAAI. 2020.

---

### Official Review · Reviewer_RhpX · 2021-07-15

**Rating:** 8
**Confidence:** 4

**Summary:**

This paper advocates a novel setting of cross-dataset and cross-domain federated learning for graph classification, which allows multiple data owners with graphs of non-iid structures and features to collaboratively train powerful graph classifiers without direct data sharing. The proposed techniques are neat yet well-justified. Extensive experiment results show clear performance gains of the proposed GCFL+ over vanilla baselines like FedAvg across different settings. Convergence analyses are also provided.

**Limitations And Societal Impact:**

Please see Main Review.

**Main Review:**

Some minor concerns are as follows:

Q1: It looks like the objective of GCFL is closer to personalized federated learning instead of standard federated learning. Can the authors clarify on this?

Q2: Although simplicity is good, the overall technical novelty is somewhat limited. For example, how is the proposed framework different from recent works such as FedCG (Cluster-driven Graph Federated Learning over Multiple Domains), in CVPR’21 workshop?

Q3: The columns in tables 3&4 can use more explanations, such as the min gain and ratio columns. As they appear now, it requires more thinking to exactly appreciate how they show the advantages of the proposed methods.

Q4: How do hyper-parameters like \epsilon_1 and \epsilon_2 influence the performance of the framework?



**Time Spent Reviewing:**

5 hours

---

> ### Author Response · Authors · 2021-08-10
> **Initial response to Reviewer RhpX**
>
> We thank the reviewer for the comments. Our responses to the several questions are as below.
>
> **Q1.** Our setting of clustered FL is indeed between personalized FL and traditional FL. Different from personalized FL, our framework allows a group of clients to share one model, so that clients have no need to maintain their own model which can be more efficient. We will add discussions about this in our problem settings in a revision.
>
> **Q2.** Our problem setting is new and the identification of its advantages itself is non-trivial. We proposed this problem and started from a basic model. We provided an improved variant in our paper, and we also shed light on the more points that can be further improved. The recent work FedCG [1] has a similar idea to ours regarding the consideration of statistical heterogeneity in datasets across domains. But the data samples they consider are images, where the statistical heterogeneity is different from the heterogeneity between graphs (graph non-iidness) as we consider. Then [1] leveraged a GNN to model the interaction between domains, which is essentially different from us in leveraging the GNN to model the data samples of graphs themselves. We will add discussions about this in our related works in a revision.
>
> **Q3.** We will refine the texts around Tables 3&4 to highlight such results regarding the unique metrics in our federated graph classification setting, to help with the understanding of the benefits of our proposed methods and facilitate future experimental studies.
>
> **Q4.** The hyper-parameter $\varepsilon_1$ is a threshold for checking whether the first general FL stage is near the stationary point and can be stopped. Theoretically, $\varepsilon_1$ should be set as small as possible. $\varepsilon_2$ is more dependent on the number of clients and the heterogeneity among them, and a smaller $\varepsilon_2$ will make the clients more likely to be clustered. The original CFL framework suggested the suitable ranges for $\varepsilon_1$ and $\varepsilon_2$. When they are in the feasible ranges, the small variation of them won’t influence the performance much since the clustering results would largely remain the same. When $\varepsilon_2$ is set too large, the performance will be similar as applying a basic FL algorithm directly (i.e. with a single cluster). If $\varepsilon_2$ is set too small, there will be more clusters with smaller sizes or even single clients. In the following, we have provided additional experimental results regarding the performance of GCFL+ regarding varying $\varepsilon_1$ and $ \varepsilon_2$. The results show that the performance of GCFL+ is not very sensitive to the values of these two hyper-parameters in pretty wide ranges.
>
> | $\varepsilon_1=0.06, \varepsilon_2=$ | 0.1 | 0.2 | 0.3 | 0.4 | 0.5 | $\varepsilon_2=0.3, \varepsilon_1=$ | 0.01 | 0.03 | 0.12 | 0.3 |
> |---|:---:|:---:|:---:|:---:|:---:|---|:---:|:---:|:---:|:---:|
> | GCFL | 0.6653 | 0.6659 | 0.6526 | 0.6499 | 0.6417 |  | 0.6290 | 0.6310 | 0.6628 | 0.6626 |
> | GCFL+ | 0.6663 | 0.6656 | 0.6585 | 0.6541 | 0.6451 |  | 0.6395 | 0.6314 | 0.6640 | 0.6663 |

---

> > ### Comment · Reviewer_RhpX · 2021-08-23
> > **Increase score to 8.**
> >
> > Thanks for the clarifications and additional results.
> >
> > After reading the authors’ responses, most of my concerns have been addressed. In my opinion, I feel this is a solid paper and I am willing to increase my score to 8.

---

> > > ### Author Response · Authors · 2021-08-23
> > > **Thanks for acknowledging our response**
> > >
> > > Dear Reviewer RhpX,
> > >
> > > We are glad to receive the positive feedback from you. Thanks a lot and please feel free to let us know should any additional concerns arise.

---

### Official Review · Reviewer_eL7Z · 2021-07-16

**Rating:** 8
**Confidence:** 4

**Summary:**

This paper provides a timely study on the emerging domain of graph federated learning, focusing on the graph classification setting. This is a natural setting close to federated learning over other types of data in terms of data split, yet the authors manage to motivate the need of doing so based on real network data analysis, and figure out the unique challenges of graphs regarding the non-iid nature of both structures and features. This provides a good motivation to the usage of GNNs, which is supported by some theoretical analysis. The existing technique of clustered FL is adopted to the GNNs and slightly improved for better performance.

**Limitations And Societal Impact:**

See main review

**Main Review:**

This paper does not try to sell a bunch of fancy or complicated architectures. Instead, it tells a real scientific story, where every major step including the problem setting, usage of GNN, utilization of clustered FL (and the small improvement) are well motivated and justified through real data analysis and observations. This has been taken for granted in recent works especially related to graph federated learning. In terms of weaknesses, I find certain low-level technical details to be confusing, such as why agglomerative clustering is used for splitting the clusters? Is it following the design of existing clustered FL ?  Is it really ideal?

Overall, this paper is well-organized and easy to follow. It provides enough information even for non-expert readers.

This paper properly justifies the need of graph classification with federated learning and demonstrates its benefit in different settings (one-dataset, cross-dataset, cross-domain). An easy-to-implement framework composed of GNN and clustered FL is proposed and shown to be effective. Though there exist some issues such as model efficiency, privacy, etc. that should be further studied, this work can serve as a good starting point for research on graph federated learning.


**Time Spent Reviewing:**

10

---

> ### Author Response · Authors · 2021-08-10
> **Initial response to Reviewer eL7Z**
>
> We thank the reviewer for the comments. Regarding the question about the clustering method used in our paper, our response is as below.
>
> The agglomerative clustering method is adapted from the original CFL framework. However, we also noticed that it is not ideally reasonable and efficient in the situation that clients are bi-clustered at each time. A cut method should be more appropriate. We just came up with a min-cut bi-section method for clustering clients. We will try to refine the clustering step with a min-cut method and update the results in a revision.

---

### Decision · Program_Chairs · 2021-09-27

**Decision:**

Accept (Poster)

**Comment:**

There was a fruitful discussion about this paper.
Overall, I feel that the paper contains enough interesting and novel ideas for a publication.